# Segment-then-Segment: Context-Preserving Crop-Based Segmentation for Large Biomedical Images

**DOI:** 10.3390/s23020633

**Published:** 2023-01-05

**Authors:** Marin Benčević, Yuming Qiu, Irena Galić, Aleksandra Pižurica

**Affiliations:** 1Faculty of Electrical Engineering, Computer Science and Information Technology, J. J. Strossmayer University, 31000 Osijek, Croatia; 2TELIN-GAIM, Faculty of Engineering and Architecture, Ghent University, 9000 Ghent, Belgium; 3Chongqing Institute of Green and Intelligent Technology, Chinese Academy of Sciences, Chongqing 400714, China

**Keywords:** biomedical images, convolutional neural networks, medical image segmentation, semantic segmentation

## Abstract

Medical images are often of huge size, which presents a challenge in terms of memory requirements when training machine learning models. Commonly, the images are downsampled to overcome this challenge, but this leads to a loss of information. We present a general approach for training semantic segmentation neural networks on much smaller input sizes called Segment-then-Segment. To reduce the input size, we use image crops instead of downscaling. One neural network performs the initial segmentation on a downscaled image. This segmentation is then used to take the most salient crops of the full-resolution image with the surrounding context. Each crop is segmented using a second specially trained neural network. The segmentation masks of each crop are joined to form the final output image. We evaluate our approach on multiple medical image modalities (microscopy, colonoscopy, and CT) and show that this approach greatly improves segmentation performance with small network input sizes when compared to baseline models trained on downscaled images, especially in terms of pixel-wise recall.

## 1. Introduction

Medical image segmentation is a key step in medical research, diagnosis, treatment, and surgical planning. A single 3D medical image, such as a CT or an MRI scan, can be up to hundreds of megabytes in size [1]. Two-dimensional images such as radiographs or digital specimen slides are often thousands of pixels in width and height. These large sizes of images present a challenge for deep learning methods. Training on larger images requires a larger amount of GPU memory and higher-capacity neural networks [2], limiting the maximum batch size and resulting in slower convergence and a worse gradient estimate. Commonly, medical images are downscaled as a pre-processing step for medical image segmentation. This leads to a loss of fine details that are often important for accurate segmentation and consequently to a reduced segmentation accuracy [3].

In this paper, we present a new approach to training segmentation convolutional neural networks (CNNs) on very small input sizes. Our approach, which we call *Segment-then-segment*, is based on cropping instead of downscaling. Thus, it maintains the information content in salient regions of the image. This method does not require changing the network architecture or capacity. The method we present can be used as a general preprocessing step for any kind of segmentation neural network.

Our approach uses two neural networks with small input sizes. The first network performs a rough segmentation on a uniformly downsampled input image. This rough segmentation is used to obtain salient crops of the original high-resolution image. These crops are then segmented separately by a second neural network trained on cropped images. A final high-resolution segmentation is built piece-wise from the segmentation masks of the individual crop regions. During cropping, we preserve the context of each object by adding padding to the crop region.

We show that this method leads to accurate segmentation with drastically smaller network input sizes. When compared to baseline models trained on uniformly downsampled images our method results in much better segmentation results, especially at small input sizes. This is a general approach and can be used for a variety of tasks and with different neural network architectures. All of the code for this paper is available at https://github.com/marinbenc/segment-then-segment (accessed on 29 November 2022).

### 1.1. Related Work

Our approach builds on the work of Qiu et al.  [4], which generates a dataset manually cropped to the object boundary, leading to an increase in segmentation performance. This was applied to skin lesions where it achieved a scale-unifying effect across the dataset. This work uses a neural network to predict the optimal object boundary as well as specific ways to train the fine segmentation network on cropped images. In addition, we allow for taking multiple crops on the image and later fusing them in the final segmentation, which makes the method applicable to a wider range of segmentation tasks.

In [5,6] a related approach is presented using the polar transform as a pre-processing step. The main novelty in this paper is the use of cropping as a transformation step. The rest of the approach was then adapted to better suit a cropping transformation, including bounding box augmentation and padding the bounding box. This allows our approach to be used to reduce the input size of the networks.

#### 1.1.1. Detect-then-Segment

Several recent end-to-end neural network architectures for segmentation incorporate cropping in one of their layers [7,8]. These generally use object detection to find a region of interest which is then segmented, sometimes called *detect-then-segment*. In models such as R-CNN [7] the objects are first detected and then fed into the segmentation pipeline. Mask R-CNN [8] uses object detection to extract a region of interest in the feature masks within the network. These methods effectively concentrate the network on a region of the image. However, information is still lost if the images are uniformly downsampled as a preprocessing step. In contrast, our approach allows one to use low input sizes by cropping the image before it enters the network. Compared to *detect-then-segment* approaches, cropping as a preprocessing step reduces the number of parameters while increasing the pixel-level information in the salient regions of the image. In addition, our approach leads to rescaling each object to the same size before the fine segmentation, which increases the scale-invariance of the models.

#### 1.1.2. Coarse-to-Fine Segmentation

Our approach can be described as a coarse-to-fine approach to image segmentation. There have been similar approaches to medical image segmentation. Zhou et al.  [9] describe an approach to pancreas segmentation using a fixed-point model [10]. They train a coarse and a fine segmentation network. The coarse network obtains an initial region of the pancreas which is then re-segmented using the fine network. The fine network then re-segments its output again, and this process is repeated iteratively until a stable solution emerges. They also use bounding box augmentation during training. Our approach differs in two ways. Firstly, we only use one iteration at a stable input size, improving the inference time. Secondly, our approach supports segmenting multiple objects on the image.

Zhu et al. [11] describe an approach to pancreas segmentation with two neural networks. The first network is a coarse segmentation network trained on overlapping 3D regions of the whole CT volume. The second network is a fine segmentation network that is trained on only the regions where the ground-truth images contain the pancreas. During inference, the fine network re-segments densely overlapping regions of the rough segmentation. The main difference in our approach is the use of only one region of interest per object where the whole object is visible and uniform in scale. This allows us to use networks of a lower capacity while still maintaining good segmentation results.

Similarly to our approach, Jha et al. [12] split the segmentation process into detection and segmentation stages. They use a neural network to first detect an object in a downsampled image. They then use the bounding box to crop the object in the high-resolution image. Our approach differs in several ways. Firstly, our approach allows the detection of multiple objects on the image and describes a way to fuse the segmentations together. Secondly, we present new ways to train the fine segmentation network to make the fine segmentation network more robust to imperfect bounding boxes. Finally, we propose a generalized approach evaluated on a variety of different modalities of medical images.

#### 1.1.3. Non-Uniform Downsampling

The resolution of an input image for neural networks can be reduced using a more complex sampling strategy. Marin et al.  [13] use non-uniform downsampling for this task. Their approach consists of training a neural network to sample points near object boundaries. The sampling points are then used to downsample the image and perform a final segmentation using a second neural network trained on the downsampled images. Similarly, Jin et al.  [14] use a learnable deformable downsampling module which is trained together with a segmentation module end-to-end. Our approach differs in the use of cropping instead of non-uniform downsampling, which preserves the topology of the image and provides localization of the object.

#### 1.1.4. Other Approaches to Reducing Input Resolution

Recently, transformer-based architectures such as SegFormer [15] and Swin Transformer [16] have become popular approaches to semantic segmentation. These networks are trained on a large number of small, overlapping patches of the image. The network uses self-attention to determine the saliency of each patch. In a sense, this allows the network to be trained on very small input image dimensions. However, transformers require a very large amount of data to be trained and have large memory requirements [17], so their use is currently limited in the domain of training on downscaled medical images.

For whole slide images, the input size is often reduced by dividing the image into equally sized patches [18,19]. A downside of this approach is computational complexity during inference since not all patches are relevant. Additionally, errors can arise when the objects are split by the patch boundary.

## 2. Materials and Methods

A visual summary of our approach is shown in Figure 1. In segmentation CNNs, commonly the image is uniformly downsampled and then segmented. This reduces the amount of information available to the network since salient and non-salient pixels are equally downsampled. Instead of uniformly downsampling the image, we crop each object in the original high-resolution image and segment the cropped images separately. The crop regions are usually much smaller than the whole image, leading to a reduction in the network input size without losing pixel information inside the objects.

The inference procedure using our method is as follows. Let *I* be an input image of size W×H. We obtain a rough segmentation using $u=g_{\phi_{1}}\left(C\left(I\right)\right)$, where *u* is a binary segmentation mask generated by $g_{\phi_{1}}$, a CNN with input and output size S×S, parameterized by $\phi_{1}$; and *C* is a uniform downsampling operation from W×H to S×S.

Given *N* connected components of *u*, a set of bounding boxes $\left\{b_{i}\right\}$ for i=1,…,N is calculated enclosing each connected component, as described in Section 2.1. Each bounding box is used to generate a crop $I_{i}=I\left(T_{i}\left(b_{i}\right)\right)$, where $T_{i}$ is a scaling and translation of the bounding box in S×S space to the corresponding region in the W×H space. Each crop is used to generate a fine segmentation mask $Y_{fi}$ using
(1)$$Y_{fi}=g_{\phi_{2}}\left(C_{i}\left(I_{i}\right)\right),$$
where $g_{\phi_{2}}$ is a CNN of the same architecture as $g_{\phi_{1}}$ and size S×S, parameterized by $\phi_{2}$, and $C_{i}$ is a scaling operation from the width and height of $I_{i}$ to S×S. A final segmentation *y* is formed using
(2)$$y=\max\{\left(T_{i}\circ C_{i}^{-1}\right)\left(Y_{fi}\right):i=1,\ldots ,N\},$$
where *y* is the resulting segmentation formed as the maximum value in all of the fine segmentations, transformed to their corresponding regions in W×H space. This process is described in more detail in Algorithm 1.

The rough segmentation network $g_{\phi_{1}}$ is trained on uniformly downsampled images. The network outputs a rough, low-resolution segmentation mask. The rough segmentation mask contains a number of connected components. For each connected component, we calculate a bounding box that encompasses all of its pixels. These bounding boxes are the crop regions used for the fine segmentation network. Since we only use this segmentation to obtain rough regions of interest, the input images to this network can be heavily downsampled without impacting the final fine segmentation.

The fine segmentation network $g_{\phi_{2}}$ is trained on cropped images using the ground truth segmentation masks to generate the bounding boxes. This network produces a fine segmentation of that region of the image. Since we know the original bounding box of each crop, we can resize the final segmentation to its original size and translate it to its original position. We perform this for each object in the image, fusing each of the fine segmentation masks into a final segmentation mask in the original image resolution.

In other words, our method performs zooming and panning around the original image and builds a final segmentation piece-wise from each zoom and pan. This allows us to use neural networks with very low input sizes without requiring a large amount of downscaling. What follows is a detailed description of different parts of the *segment-then-segment* process.

**Algorithm 1** Inference algorithm for one input image**Input:** High-resolution input image *I* of size H×W, input size *S*, padding *k*, neural network NET_1 trained in S×S downscaled images, neural network NET_2 trained on ground truth S×S image crops.**Output:** Output image *Y* of size H×W. I′←RESIZE(I,(S,S)) y′←NET_1(I′) y′←RESIZE(y′,(H,W)) ccs←CONNECTED_COMPONENTS(y′) crops←[]                       ▹ An array of S×S images bboxes←[]                 ▹ An array of bounding boxes for each crop**for** cc in ccs **do**     bbox←bounding_box(cc)     bbox.width←bbox.height←max(bbox.width,bbox.height)     bbox←(bbox.left−k,bbox.top−k,bbox.width+k,bbox.height+k)     bbox←SHIFT_TO_IMAGE_REGION(bbox,(H,W))     crops.ADD(CROP(I,bbox))     bboxes.ADD(bbox)**end for****for** crop, i in crops **do**     l,t,w,h←bboxes[i]     crop←RESIZE(crop,(S,S))     ycrop←NET_2(crop)     ycrop←RESIZE(ycrop,(h,w))     Y[t:t+h,l:l+w]←Y[t:t+h,l:l+w]||ycrop **end for**

### 2.1. Cropping

The key to reducing downscaling in our approach is that we take crops from the image in the original resolution. The crop regions themselves are predicted on a downscaled image and then projected to the original image space.

The cropping procedure is as follows. First, a bounding box fully encompassing each connected component in the rough segmentation is calculated. The coordinates of the bounding box are then scaled to the high-resolution image space. An empirically determined padding of S/8 (for an S×S input size) is added along each of the four sides. This way of cropping preserves the context of the object and decreases the number of false negatives inside the bounding box.

The box is then squared, i.e., its height and width are set to the larger of the two dimensions. The box is also shifted (maintaining its width and height) to be fully inside the region of the image. Finally, the bounding box is used as a region to crop the original high-resolution image. The rough segmentation sometimes results in noisy regions, so each crop whose width or height is less than 5 pixels is discarded.

### 2.2. Fine Segmentation and Fusion

Each crop of the high-resolution image is scaled to the input size of the fine segmentation network. The fine segmentation network outputs a number of segmentation masks equal to the number of connected components in the rough segmentation. A high-resolution segmentation mask is created by translating and scaling each of the fine segmentation network outputs to their original position. By doing so we construct a full segmentation piece by piece, where each piece is a fine segmentation of a single object in the image, as detected by the rough segmentation. If the cropped regions overlap in the final segmentation, we use a logical OR operator to resolve the conflict in the fine segmentation mask. This process is presented in Algorithm 1.

### 2.3. Training the Fine Segmentation Network

The fine segmentation network is trained on ground truth image crops. The crops are obtained by using the connected components of ground truth segmentation masks. From there, the crops are prepared in the same way as described in Section 2.1. Since the images have multiple crop regions, we choose one of the crop regions of the full-resolution image at random during each training iteration. If the original image has no connected components in the ground truth segmentation mask, the whole image is used as training input. All input images to the fine segmentation network are resized to S×S, where *S* is a pre-determined input size that matches the input size used to train the rough segmentation network.

During training, we add an additional augmentation step in the form of crop region jittering. When preparing the crops during training, a uniformly distributed random number between ±16 pixels is added to each dimension of the bounding box (the *x*- and *y*-coordinate, width, and height). This ensures that the trained network is robust to imperfect rough segmentation masks during inference.

In our experiments, we used the same architecture for both the rough and fine segmentation networks, as this allows us to use transfer learning. However, there is no requirement that the networks use the same architecture.

## 3. Results

We evaluate our approach on three separate datasets described in detail in Section 3.1, hereafter referred to as the cells, aorta, and polyp datasets. First, for each dataset, we trained a rough segmentation U-Net, Res-U-Net++, and DeepLabv3+ network at various downscaled input resolutions. These models are also used as baseline models to compare against our approach. To evaluate our approach, we train fine segmentation models using the same combinations of datasets, architectures, and input sizes.

Altogether more than 100 neural networks were trained to evaluate our approach, including both the baseline networks and networks trained on cropped images.

Each network is trained from scratch using the downscaled dataset. The outputs from the networks are then upscaled to the datasets’ original resolution, and the metrics are calculated using those outputs. We use the held-out test datasets for all of the results reported in this section. The baseline models are used as the rough segmentation networks for the experiments using our approach. The hyperparameters used for each network are reported in Table 1.

In the interest of providing objective metrics of model performance, all of the hyperparameters were tuned using the validation dataset on the baseline U-Net. Those same hyperparameters are then used for each of the models in our approach. Each model is trained using the Adam optimizer up to a maximum number of epochs and the best model with the best validation loss is saved during training. The validation loss is calculated as the Dice score coefficient (DSC) over the validation dataset at the same resolution as the input images. We do not upscale the outputs for the validation loss as we do for calculating the final metrics. Each model was trained using PyTorch 1.10 on an Nvidia GeForce RTX 3080 GPU. Where possible, we have fixed the random seed value to “2022”, but we have also run the experiments on two other random seeds and obtained similar results.

### 3.1. Datasets

This section briefly describes the datasets used in our experiments as well as the preprocessing steps for the images. For more details, we direct readers to the supplemental code repository available at https://github.com/marinbenc/segment-then-segment (accessed on 29 November 2022). To evaluate our approach, we chose three datasets across different medical imaging modalities, including CT scans, microscopy imaging, and colonoscopy images. We hope that the variety in the datasets will show the generalizability of our approach. Aside from the variety, the datasets were selected because they include images of large dimensions on which small objects of various sizes need to be segmented. These types of tasks are most likely to suffer from the loss of information due to downscaling and are thus particularly suitable to be segmented using our approach.

#### 3.1.1. Aorta Dataset

For aorta segmentation we use the AVT dataset [20], a multi-center dataset of labeled CTA scans of the aortic vessel tree. We only use a subset of the dataset from Dongyang Hospital, a total of 18 scans of between 122 and 251 slices. Each slice is windowed to 200 to 500 HU, normalized to [−0.5,0.5], and zero-centered by subtracting 0.1 from each slice. The original resolution of the slices is 512×666 pixels. We use augmentation during training. Each input has a 50% chance of an affine transform (translation of ±6.25%, scaling of ±10%, rotation of $\pm 14^{\circ }$), as well as a 30% chance of a horizontal flip. The dataset is split per patient into a training set (70%, 39 scans, 14,147 slices), validation set (20%, 11 scans, 4488 slices), and test set (10%, 6 scans, 3119 slices).

#### 3.1.2. Cells Dataset

For cell nucleus segmentation we use the 2018 Data Science Bowl dataset, otherwise known as image set BBBC038v1 from the Broad Bioimage Benchmark Collection [21]. We use 670 RGB images and their corresponding labels from the stage1_train repository. The original files are of various sizes ranging from 256×256 to 1024×1024 pixels. We did not further split the images into patches, all training is done on the whole images. We use the same augmentation as described in Section 3.1.1. The dataset is split into a training set (80%, 536 images), validation set (10%, 67 images), and test set (10%, 67 images).

#### 3.1.3. Polyp Dataset

For polyp segmentation, we use the Kvasir-SEG dataset [22], which contains 1000 annotated gastroscopy images containing polyps. The size of the original images ranges from 332 to 1920 pixels in width. We split the dataset into train (80%, 800 images), validation (10%, 100 images), and test (10%, 100 images) datasets.

### 3.2. Quantitative Assessment

The results of our experiments are shown in Table 2. Our approach, using low input sizes, results in segmentations on par or better than baseline models trained on larger input sizes using downscaled images. This is especially apparent at large downscaling factors of 4 and more, where the baseline models quickly deteriorate in their performance but our models were still able to achieve results that are close to those on full-size images. The biggest improvement is seen in terms of recall. For instance, on the 4× downscaled cells dataset, our approach leads to a 4.3 times larger recall. Likewise, for the polyp images, the recall improves from 0.039 to 0.799, a 20 times improvement. Recall is especially important in medical image segmentation since the cost of false negatives can often far outweigh the cost of false positives.

We evaluate how general our approach is by applying it to two other state-of-the-art semantic segmentation architectures, Res-U-Net++ [23,24] and DeepLabv3+ [25]. The results of these experiments are shown in Table 3.

Furthermore, our approach achieves much better stability of results as the input size decreases. This is shown visually in Figure 2. The stability improvements are especially visible in Figure 3, where it can be seen that the distribution of the results from our approach remains more stable than in the baseline models.

The goal of our approach is increasing segmentation performance on downscaled images, so we do not expect a performance increase on the full-size images. While our approach offers significant improvements when using downscaled images, the main disadvantage of our approach is that it requires training two separate neural networks. However, this downside is lessened by two key factors. First, the cropped networks converge much faster since the objects are already localized and unified in scale in the images, making the problem easier for the network to learn. Secondly, since the architecture of the two networks is identical, one can use transfer learning from the trained rough segmentation network to the fine segmentation network.

#### 3.2.1. Qualitative Assessment

Qualitatively, there is a large improvement when using our approach over the baseline methods on downscaled inputs. At large downscaling factors, outputs from the baseline models often include artifacts on the border since the pixel grid is not fine enough to represent small variations on the object boundary. These issues disappear with our approach, as the overall downscaling amount is much lower. This effect is especially visible on the cells dataset due to its relatively small object size, as shown in Figure 4a. We observe a similar effect on the aorta dataset, shown in Figure 4c.

On the polyp dataset, our approach greatly reduces the number of false negative pixels on the image, as can be seen in Figure 4b. U-Net fails to predict the whole polyp region on small input sizes, leading to ragged object boundaries and holes in the predicted regions. By comparison, our approach produces smoother, closed contours and manages to capture more of the object boundary than U-Net equivalents at the same input size.

#### 3.2.2. Computational Performance Characteristics

Since our approach consists of using a cascade of two U-Nets, the number of parameters of the network is doubled. However, the peak GPU memory utilization during training and inference remains exactly the same when using our approach as with a baseline model on the same input size. Our approach allows one to reduce the input size while still maintaining the same segmentation metrics, thus allowing larger batch sizes during training. This is presented in Table 4.

In terms of computational performance, the largest downside of our approach is that it increases inference time non-linearly with the size of the images, which can be seen in Figure 5.

This increase is explained by two key features of our approach. Firstly, cropping and rescaling operations take up a large percentage of the time. Secondly, each connected component in the initial segmentation is processed separately by the second network, thus inference time increases with the number of objects to be segmented. This is most apparent in the cells dataset, as the input images have the largest number of objects. Note that we use the implementation of torch.nn.functional.interpolate with the nearest-neighbor mode in Pytorch 1.10 for all resizing. The inference time depends on the specific implementation of the resizing algorithm as well as the hardware it is being run on.

While these increases seem large in relative terms, it should be noted that in absolute terms the increases are in the order of magnitude of tens of milliseconds. We argue that such an increase in inference time does not limit the applicability of our approach in practical use cases.

## 4. Conclusions

Downscaling is a common source of segmentation errors in neural networks. In this paper, we present an approach to training neural networks that reduces downscaling by utilizing two neural networks and salient crops. We show how training a second neural network on cropped image regions can improve segmentation performance on small input sizes with few downsides. Our approach improves segmentation metrics on downscaled images across different modalities and image sizes, especially in terms of recall. We show that, while this approach increases inference time, it allows for training using much larger batch sizes while maintaining the same segmentation metrics.

Note that the goal of our method is not to produce state-of-the-art segmentation results on high-resolution images. Instead, the goal is to allow training on heavily downscaled images without sacrificing segmentation performance.

Our approach is a general preprocessing method and can be applied to a variety of different segmentation tasks regardless of the underlying architecture, so long as the two networks output a segmentation mask. In addition, the rough segmentation portion can be substituted with any method that produces a bounding box for each object. Aside from object detection methods, the bounding boxes could also be determined manually by an expert, as shown in [4]. While we did not evaluate our approach on 3D networks in this paper, there is nothing in our approach that is specific to 2D images. Our approach can be extended to 3D images by using 3D neural network architectures and 3D bounding boxes for crop regions.

In addition, our approach could greatly benefit from using transfer learning, as the two networks use the same underlying architecture. It is possible that the results could be further improved with good transfer learning datasets as well as more complex training regimes such as contrastive learning [26]. During the development of this paper, we experimented with training the architecture end-to-end but failed to produce ways for the rough segmentation network to converge in a stable manner. We plan to explore this further in future work.

We believe that this approach will allow future researchers to train standard semantic segmentation neural networks on downscaled versions of very high-resolution images without sacrificing segmentation performance. We also show that these results are general across a variety of biomedical images and thus can be applied to a very large number of problems in this space.

## Figures and Tables

**Figure 1 sensors-23-00633-f001:**
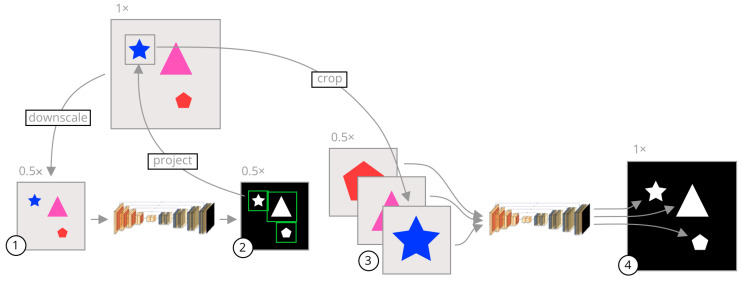
A visual summary of our approach. The gray images are the input images while the black images are segmentation mask outputs from the models. The shapes on the images are only representative, and the inputs can be any image where several objects need to be segmented. The arrows represent image operations. (1) An image is uniformly downsampled from its original resolution. (2) A rough segmentation is predicted by a neural network, and the bounding box of each connected component is calculated. (3) The bounding boxes are scaled to the original image space and crops of the input image are taken in the original resolution and scaled to a common input size. (4) Each crop is segmented separately by a second neural network specifically trained on cropped images. These crops are fused together to form a final segmentation in the original high resolution.

**Figure 2 sensors-23-00633-f002:**
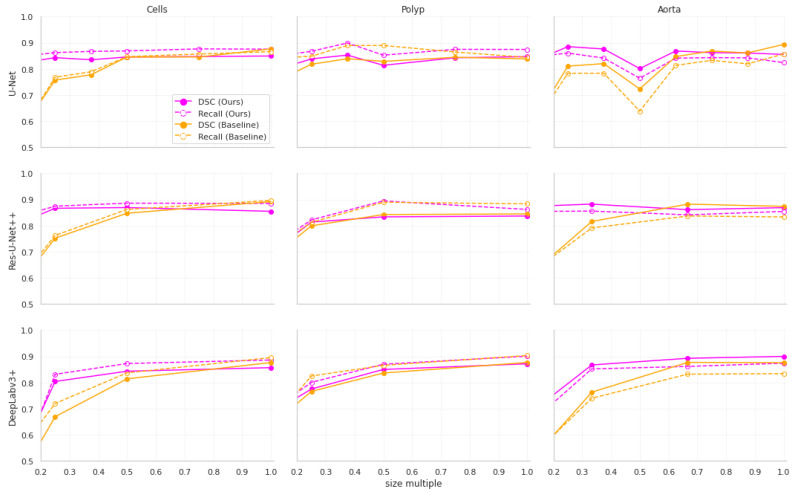
The relationship between input dimensions and the mean Dice Score Coefficient (DSC) and recall for different datasets. The points are measured values from our experiments.

**Figure 3 sensors-23-00633-f003:**
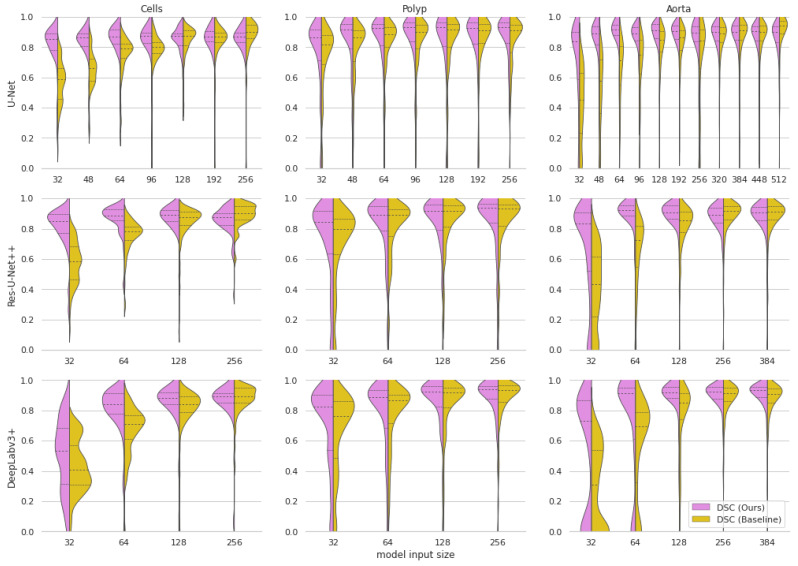
Violin plots of Dice Score Coefficients of our approach compared to the baseline models at various input dimensions. The dashed lines represent quartiles of the distributions.

**Figure 4 sensors-23-00633-f004:**
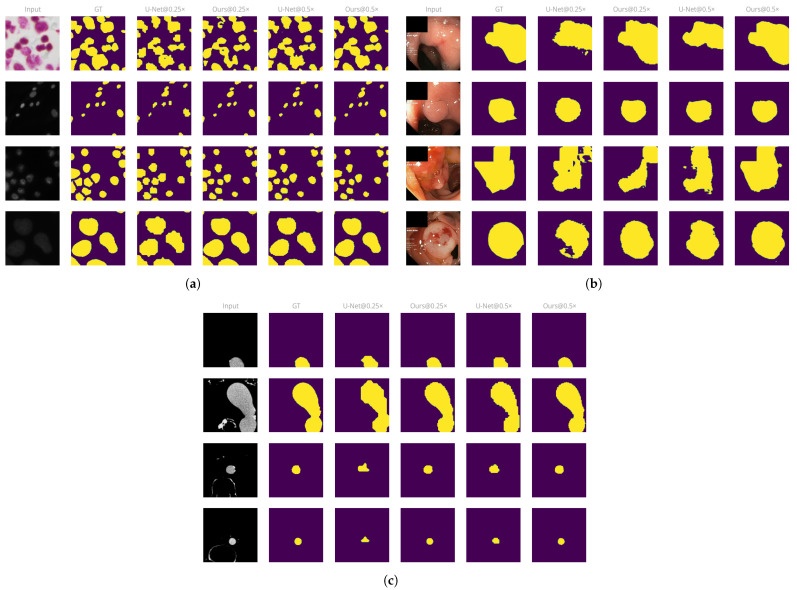
Example outputs from the models at various input sizes. (**a**) Cells dataset. (**b**) Polyp dataset. (**c**) Aorta dataset.

**Figure 5 sensors-23-00633-f005:**
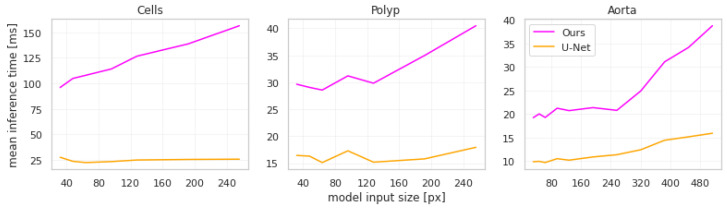
Mean per-input inference time across different input sizes for the U-Net-based models.

**Table 1 sensors-23-00633-t001:** The hyper-parameters used for each of the models in our experiments.

Dataset	Batch Size	Learning Rate	Max. Epochs
Cells	16	$5\cdot 10^{-4}$	100
Polyp	8	$10^{-3}$	175
Aorta	8	$10^{-3}$	100

**Table 2 sensors-23-00633-t002:** The results of our approach U-Net as the underlying architecture. The complete results of all of our experiments are available in Appendix A.

	DSC [%]	IoU [%]	Prec. [%]	Rec. [%]
**Cells** (256×256)				
64×64—U-Net	75.70±11.82	62.13±13.29	75.27±12.61	76.79±12.70
64×64—Seg-Then-Seg	84.28±12.83	74.50±15.45	84.75±14.85	86.22±13.25
128×128—U-Net	84.52±10.34	74.33±13.05	85.21±11.61	84.66±11.61
128×128—Seg-Then-Seg	84.54±9.64	74.26±12.73	83.97±13.00	86.86±9.39
256×256—U-Net	87.62±13.55	79.75±15.46	89.44±14.09	86.58±14.83
256×256—Seg-Then-Seg	84.91±12.65	75.24±13.92	83.52±14.50	87.53±13.76
**Polyp** (256×256)				
64×64—U-Net	81.76±18.81	72.40±21.00	83.94±21.36	84.84±17.30
64×64—Seg-Then-Seg	83.82±19.91	75.90±22.51	86.32±21.87	86.72±17.84
128×128—U-Net	82.86±20.72	74.75±23.32	83.30±22.73	88.93±17.50
128×128—Seg-Then-Seg	81.36±26.57	74.58±27.52	82.49±27.55	85.23±25.33
256×256—U-Net	83.80±16.08	74.88±20.44	87.91±19.29	84.37±16.55
256×256—Seg-Then-Seg	84.68±19.40	77.11±22.53	87.10±20.91	87.42±18.41
**Aorta** (256×256)				
128×128—U-Net	81.03±14.45	70.13±17.02	85.09±14.14	78.30±15.04
128×128—Seg-Then-Seg	88.50±11.08	80.73±13.93	92.42±10.33	86.02±12.25
256×256—U-Net	72.30±28.69	62.97±28.74	91.21±25.39	63.80±29.11
256×256—Seg-Then-Seg	80.10±25.18	72.12±25.74	86.75±24.14	76.39±26.51
512×512—U-Net	89.34±14.59	83.10±18.26	96.03±11.52	85.66±17.44
512×512—Seg-Then-Seg	85.51±14.80	76.85±17.17	90.90±14.15	82.39±15.54

**Table 3 sensors-23-00633-t003:** A comparison of the Dice Score Coefficients of our approach using other underlying architectures at 4× and 2× downscaled images. The complete results of all of our experiments are available in Appendix A.

Model Cells (256×256)	Size	Baseline [%]	Seg-then-Seg [%]
Res-U-Net++	64×64	75.18±11.85	86.63±10.06
Res-U-Net++	128×128	84.74±9.64	86.90±11.77
DeepLabv3+	64×64	66.93±15.93	80.41±16.80
DeepLabv3+	128×128	81.43±13.74	84.34±16.91
**Polyp** (256×256)			
Res-U-Net++	64×64	79.92±19.72	81.35±20.82
Res-U-Net++	128×128	84.23±17.74	83.34±20.73
DeepLabv3+	64×64	76.62±21.78	77.53±24.56
DeepLabv3+	128×128	83.65±18.86	85.02±19.37
**Aorta** (512×512)			
Res-U-Net++	128×128	81.59±14.43	88.21±13.25
Res-U-Net++	256×256	88.21±13.19	86.11±14.02
DeepLabv3+	128×128	76.30±22.81	86.73±16.94
DeepLabv3+	256×256	87.71±12.13	89.28±11.75

**Table 4 sensors-23-00633-t004:** Performance characteristics of our approach compared to the baseline model with similar mean test Dice Score Coefficients.

	Input Size	DSC	Peak VRAM ${}^{1,3}$	Inf. Time ${}^{2,3}$	Max. Batch Size ${}^{3}$
**Cells**					
U-Net	$64^{2}$	75.70	1.9 GB	24 ms	300
Seg-Then-Seg	$64^{2}$	84.28	1.9 GB	122 ms	300
U-Net	$128^{2}$	84.52	2.3 GB	24 ms	80
**Polyp**					
U-Net	$48^{2}$	78.42	2.9 GB	16 ms	850
Seg-Then-Seg	$48^{2}$	81.68	2.9 GB	29 ms	850
U-Net	$128^{2}$	82.86	3.8 GB	16 ms	150
**Aorta**					
U-Net	$128^{2}$	81.03	2.9 GB	13 ms	46
Seg-Then-Seg	$128^{2}$	88.50	2.9 GB	26 ms	46
U-Net	$512^{2}$	89.34	10.1 GB	16 ms	9

^1^ Measured using a batch size of 8 for all rows. ^2^ Mean inference time across all inputs in the test set, calculated per slice for the aorta dataset. ^3^ Measured using PyTorch 1.10 on an Nvidia GeForce RTX 3080 GPU and an AMD Ryzen 7 3700× 8-core CPU with 32 GB of RAM.

## Data Availability

Not applicable.

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
