# Peer review of "Segment-then-Segment: Context-Preserving Crop-Based Segmentation for Large Biomedical Images"

_sensors, 2023, doi:10.3390/s23020633_

Round 1

Reviewer 1 Report

1. What’s the difference between the segment-then-segment method in this paper and the SOTA (coarse-to-fine/detect-then-segment) methods?

2. Line 64 “Another way to tackle this challenge...”. What is the challenge referred to does not state in the related works?

3. What is the difference between the author’s approach and reference 4? what improvements have been made to make it more general? Or just evaluated on a variety of different modalities of medical images?

4. Figure 1, is the image resolution (256*256) of step 3 wrong? Does it a fixed value, or does it change with the size of the bounding box? Besides, if the images are noted using spatial resolution, it could be easier for the reader to understand, such as the original image, step 3 and 4 labeled as (2x, 2x), and step 1 and 2 for (x, x).

5. Figure 1, it is said that “These crops are fused to form a final segmentation in the original high resolution.” How these crops are fused? Since each crop image is segmented separately, how to get the spatial location information of each crop and fuse them?

6. The authors did not compare their approach with other coarse-to-fine/detect-then-segment methods. Is there any performance improvement compared to the SOTA methods?

7. Does “downscaled” and “downsampled” the same thing all around the text?

8. The language used throughout this paper needs to be improved, the author should do some proofreading on it.

Author Response

First of all, we would like to thank the reviewer for the helpful comments.

1) “What’s the difference between the segment-then-segment method in this paper and the SOTA (coarse-to-fine/detect-then-segment) methods?”

Coarse-to-fine and detect-then-segment methods (such as Mask R-CNN) effectively concentrate the network on some region of the image. However, information is still lost if the images are uniformly downsampled as a preprocessing step. In contrast, our approach allows one to use lower-resolution inputs to the network by cropping and not downscaling. Compared to detect-then-segment and coarse-to-fine approaches, our method reduces the number of parameters of the network while increasing the pixel-level information in the salient regions of the image. In addition, our approach leads to rescaling each object to the same size before the fine segmentation, increasing the models' scale invariance.

We agree that this was not clearly stated, and this has been clarified and revised in Section 1.1.

2) “Line 64 “Another way to tackle this challenge...”. What is the challenge referred to does not state in the related works?”

This phrase has been completely rewritten in the paper as it was unclear.

3) “What is the difference between the author’s approach and reference 4? what improvements have been made to make it more general? Or just evaluated on a variety of different modalities of medical images?”

An important difference between the approach presented by Jha et al. and our approach is that our approach allows for the detection of multiple objects in the image. We do so by extracting each connected component in the image and segmenting it separately and then joining the resulting segmentations. In addition, we present new ways to train the fine segmentation network such as e.g. bounding box augmentation and selecting a random bounding box during training to make the fine segmentation network more robust to imperfect bounding boxes.

We have clarified this in more detail in Section 1.1. of the paper.

4) “Figure 1, is the image resolution (256*256) of step 3 wrong? Does it a fixed value, or does it change with the size of the bounding box? Besides, if the images are noted using spatial resolution, it could be easier for the reader to understand, such as the original image, step 3 and 4 labeled as (2x, 2x), and step 1 and 2 for (x, x).”

After salient crops are taken in the original high-resolution image, they all need to be scaled to a unified size so that they can be segmented by the network. This results in scaling (either up or down depending on the size of the crop), but the total scaling is much lower than if the whole image were to be scaled. We have clarified this in the caption of Figure 1. Also, we followed the suggestion of the reviewer to improve the clarity of the notation by using “1x” and “0.5x” to denote the scale of each image.

5) “Figure 1, it is said that “These crops are fused to form a final segmentation in the original high resolution.” How these crops are fused? Since each crop image is segmented separately, how to get the spatial location information of each crop and fuse them?”

The bounding box used to obtain each crop is saved before it is segmented, and is then used to place the fine segmentation map of the crop to the correct position in the final fused segmentation map. While the caption only mentions that the images are “fused” to avoid too much repetition, we have more clearly explained this process in Section 2.2 and changed the title of the section to “Fine Segmentation and Fusion” to make it easier to find.

6) “The authors did not compare their approach with other coarse-to-fine/detect-then-segment methods. Is there any performance improvement compared to the SOTA methods?”

The main goal of our approach is to perform segmentation with neural networks of low input sizes without sacrificing segmentation quality. In other words, we aim to improve the tradeoff between input size and segmentation performance. To this end, we compare our approach to baseline models at various input sizes to show how our approach maintains segmentation performance as the input resolution reduces.

We would like to stress that our approach is not a novel segmentation model but a preprocessing step, which is compatible with coarse-to-fine as well as detect-then-segment approaches.

7) “Does “downscaled” and “downsampled” the same thing all around the text?”

The meaning of the two terms in the related literature and also in this manuscript is not exactly the same in the sense that the term “downscaling” is more general. By downscaling we mean reducing the image size irrespective of the particular approach. Downsampling refers to sampling an image at a lower rate, which consequently results in a reduced image size.

When use “downsampling” when mentioning specific methods that use sampling such as uniform downsampling and “downscaling” when referring to the general process of scaling an image.

8) “The language used throughout this paper needs to be improved, the author should do some proofreading on it.”

We have made several extensive revisions to the paper. We clarified, rewrote, and edited large portions of the paper to address this concern. We hope that the language is clearer and more concise in this revision.

Reviewer 2 Report

The authors present a general preprocessing method for medical images in order to improve segmentation.

The paper need extensive edition on English language, grammar and style. There are a lot of very log phrases, with wrong topic. Some phrases are hard to be followed. Also, punctuation should be revised.

Preprocessing step is a common step in image processing. There are various methods used to downsize images. One popular method is based on wavelets and the wavelet transform. Did you consider trying the wavelets theory for this purpose (even combined with neural networks)?

Author Response

Firstly, we would like to thank the reviewer for the insightful comments and suggestions.

Here is our response to each of the points of the review:

  1. The paper need extensive edition on English language, grammar and style. There are a lot of very log phrases, with wrong topic. Some phrases are hard to be followed. Also, punctuation should be revised.”

We have rewritten large portions of the manuscript and significantly improved the language use and the writing style. The manuscript has been carefully proofread to correct any mistakes.

  1. “Preprocessing step is a common step in image processing. There are various methods used to downsize images. One popular method is based on wavelets and the wavelet transform. Did you consider trying the wavelets theory for this purpose (even combined with neural networks)?”

The wavelet transform can indeed be utilized to increase the receptive fields of neural networks without sacrificing performance and a lot of research has been done on the subject. In terms of preprocessing, wavelets can reduce the amount of non-salient information. This has been shown to be beneficial for different kinds of medical imaging tasks and is an interesting area of research.

However, our paper aims to improve the tradeoff between segmentation performance and input size with no changes to the network architecture. This allows the use of common state-of-the-art architectures as well as taking advantage of transfer learning from previously trained models using the same architectures. This is why we have opted for using image crops, as it is a very general transformation compatible with most convolutional neural networks.

Round 2

Reviewer 1 Report

None.

Reviewer 2 Report

The authors have followed my observations and have modified the paper accordingly. So, I don't have further observations.